# A Boolean approach for novel hypoxia-related gene discovery

Tsering Stobdan[1], Debashis Sahoo[1,2], Gabriel G. Haddad [1,3,4]*

**1** Department of Pediatrics, Division of Respiratory Medicine, University of California San Diego, La Jolla, California, United States of America, **2** Department of Computer Science and Engineering, University of California San Diego, La Jolla, California, United States of America, **3** Department of Neurosciences, University of California San Diego, La Jolla, California, United States of America, **4** Rady Children's Hospital, San Diego, California, United States of America

* ghaddad@health.ucsd.edu

## Abstract

Hypoxia plays a major role in the etiology and pathogenesis of most of the leading causes of morbidity and mortality, whether cardiovascular diseases, cancer, respiratory diseases or stroke. Despite active research on hypoxia-signaling pathways, the understanding of regulatory mechanisms, especially in specific tissues, still remain elusive. With the accessibility of thousands of potentially diverse genomic datasets, computational methods are utilized to generate new hypotheses. Here we utilized Boolean implication relationship, a powerful method to probe symmetrically and asymmetrically related genes, to identify novel hypoxia related genes. We used a well-known hypoxia-responsive gene, *VEGFA*, with very large human expression datasets (n = 25,955) to identify novel hypoxia-responsive candidate gene/s. Further, we utilized *in-vitro* analysis using human endothelial cells exposed to 1% $O_2$ environment for 2, 8, 24 and 48 hours to validate top candidate genes. Out of the top candidate genes (n = 19), 84% genes were previously reported as hypoxia related, validating our results. However, we identified *FAM114A1* as a novel candidate gene significantly upregulated in the endothelial cells at 8, 24 and 48 hours of 1% $O_2$ environment. Additional evidence, particularly the localization of intronic miRNA and numerous HREs further support and strengthen our finding. Current results on *FAM114A1* provide an example demonstrating the utility of powerful computational methods, like Boolean implications, in playing a major role in hypothesis building and discovery.

## Introduction

Oxygen is vital to the living cells, especially critical in high-energy requiring tissues like brain, heart, liver and kidneys, and therefore it plays a dominant role in the pathogenesis and pathophysiology of most of the major diseases [1–3]. Since an impaired oxygen supply (hypoxia) is the basis for these diseases i.e., cardiac ischemia, stroke and chronic obstructive pulmonary disease (COPD), all advances in the treatments are focused on methods that could maintain a steady $O_2$ supply, little is known about treating or preserving the affected cells, especially the

**Data Availability Statement:** All relevant data are within the manuscript and its Supporting Information files or available online at http://hegemon.ucsd.edu/Tools/explore.php?key=global.

**Funding:** This work was supported by NIH grant (R01 HL127403-04) GGH.The funders had no role in

in study design, data collection and analysis, decision to publish, or preparation of the manuscript.

**Competing interests:** The authors have declared that no competing interests exist.

terminally differentiated cardiomyocytes and neurons [4, 5]. Although the identification of hypoxia inducible factor (HIF), the central regulator of hypoxia [6], and numerous other hypoxia related genes, have recognized potential therapeutic targets [7, 8], our knowledge on the molecular mechanisms that can be targeted to protect or increase the tolerance of these cells to hypoxia is primitive.

The conventional approaches like the candidate gene assessment or 'omics' approaches such as genomics, transcriptomics, proteomics and metabolomics, so far have helped us to identify critical targets [7]. However, it is clear that identification of new or next generation of targets will require data from diverse samples types and enormous sample numbers and a newer method to analyze. The emerging tools of network analysis or systems biology that utilizes data from databases that has large-scale diverse sample types, offers a platform to understand the complexity and to visualize and prioritize targets [9, 10], which addresses the current challenges to some extent. Since most of these tools are based on the symmetric relationship between a pair of genes [9, 10], overlooking the larger set of the asymmetric relationship, newer approaches are evolving to identify both symmetric and asymmetric relationships [11]. In the current study, our goal was to identify novel hypoxia-related gene using Boolean implications relationship i.e., that follow simple "if-then" rules, [11, 12]. For this we used MiDReG (mining developmentally regulated genes), a bioinformatics method initially developed that identifies developmentally regulated genes but has the potential to examine the transcriptional relationships, symmetric and asymmetric, between any known genes from thousands of microarray experiments.

## Results

### Boolean implication analysis predicts hypoxia-responsive genes

In order to identify novel hypoxia-responsive genes, we utilized more than 25,955 publicly available human dataset and perform Boolean implication analysis to filter candidate genes that have a Boolean relationship with the seed gene *VEGFA* (vascular endothelial growth factor A, Gene ID: 7422, Fig 1A) i.e., a well-established hypoxia responsive/sensitive gene. Out of the four probesets (cDNA fragment or oligonucletides that represent genes) targeting *VEGFA* i.e., 210512_s_at, 210513_s_at, 211527_x_at and 212171_x_at, we used probeset 210512_s_at in our primary analysis because of its robust signal. The software MiDReG (mining developmentally regulated genes) which couples gene expression patterns with "if-then" rules (Boolean implications) was used to predict functionally important *VEGFA*/hypoxia-related genes relationships [11, 12]. At a BooleanNet statistical threshold of SThr > 40 and pThr < 0.2 MiDReG detected 310 probesets (230 genes) that pairs with *VEGFA* (S1 Table). Out of the six potential gene relationships discovered by BooleanNet, two symmetric (Equivalent and Opposite) and four asymmetric, the 310 probesets show only five Boolean relationship and no Boolean Opposite relationship (S1 Table). For example, out of 230 genes two genes shows high = >high implication, one shows high = >low implication, three shows low = >high implication and 216 shows low = >low implication with *VEGFA*. We picked 22 probesets (19 genes) that have a strong relationship with *VEGFA* (S2 Table). Boolean relationships here include 'equivalent' for *SEPT10*, *TEAD1*, *FKBP9*; 'high = >low' for *RASGRP2*; 'low = >high' for *VAV1*, *PRKCB1*, *MFNG*, *CUGBP2*; 'low = >low' for *YAP1*, *SH3D19*, *FAM114A1*, *EMP2*, *LGR4*, *NR2F2*, *PPIC*, *ERRFI1* and high = >high for *LAPTM4B* and *ITGAV*. Interestingly, eleven genes (85%) were previously known to be hypoxia related (S2 Table).

We subsequently validated the hypoxia response of some of the candidate genes discovered, along with the seed gene, *VEGFA*, in human pulmonary artery endothelial cells (HPAECs) exposed to 1% O$_2$ environment for 2, 8, 24 and 48 hours (Methods). The results showed an

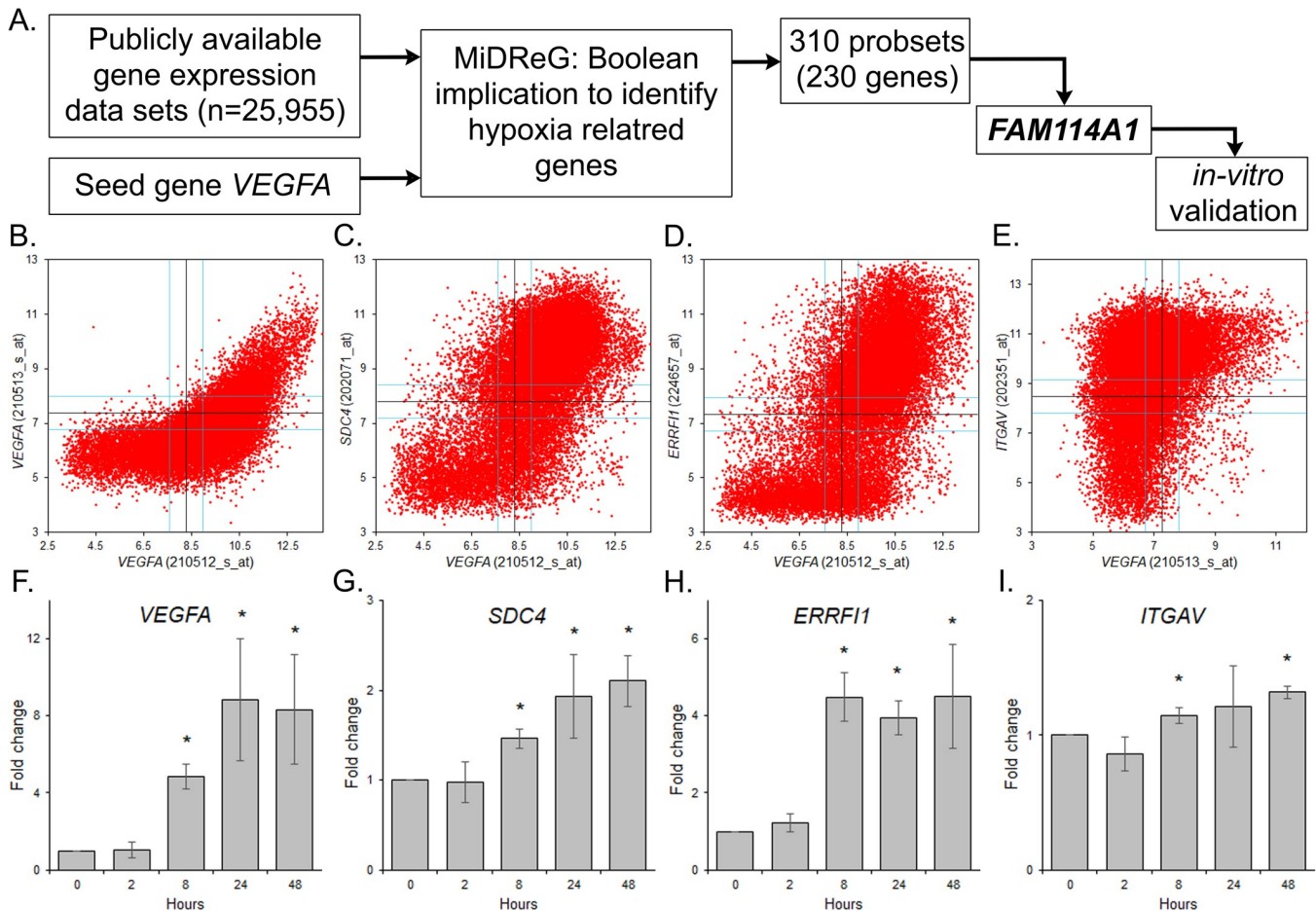

**Fig 1. Boolean implication analysis to identify novel hypoxia-responsive genes.** (a) Schematics of approach utilized to filter hypoxia-related genes. We used *VEGFA* as the seed gene to identify genes responding to hypoxia. Filter includes ranking, conserved relationship in mice and novelty. (b—e) Scatter plots depicting the expression profile of 25,955 samples for *VEGFA* (probset 210512_s_at, x-axis) and the additional *VEGFA* probset 210513_s_at (c) and top correlated genes i.e., *SDC4* (d), *ERRFI1* (e) and *ITGAV* (f). (f—i) Validation of candidate seed gene (*VEGFA*) and *SDC4* (h), *ERRFl1* (i) and *ITGAV* (j) in HPAECs in normoxia (21% $O_2$) and at 2, 8, 24 and 48 hours of constant 1% $O_2$. (*, P<0.05).

upregulation of *VEGFA* and some of the candidate genes at the 8 hours of hypoxia treatment (Fig 1). For example, the expression levels of *VEGFA* and the tested genes *SDC4*, *ERRFI1* and *ITGAV* increased significantly after 8 hours of 1% $O_2$ (Fig 1). As anticipated, the expression of *HIF1a* does not change during the course of hypoxia (S1 Fig).

## Boolean implication analysis identifies *FAM114A1* as a novel hypoxia-related gene

At BooleanNet statistical value of SThr = 48.06 and pThr = 0.08, *FAM114A1* is the top candidate with Boolean implication *VEGFA* low = > *FAM114A1* low relationship (Fig 2A, S1 Table). Furthermore, analysis of the mouse data using 11,758 publicly available samples also reveal a similar relationship i.e., *Vegfa* high = > *9130005N14Rik* high (Fig 2B). In order to validate the hypoxia sensitivity of *FAM114A1*, we measured its transcript levels in HPAECs treated with 1% $O_2$ for 2, 8, 24 and 48 hours. The expression level increases significantly at 8, 24 and 48 hours of hypoxia treatment (P<0.05; Fig 2C) which is consistent with upregulation of *VEGFA* post 8 hours of hypoxia (Fig 1F). However, the fold change in *FAM114A1* is modest,

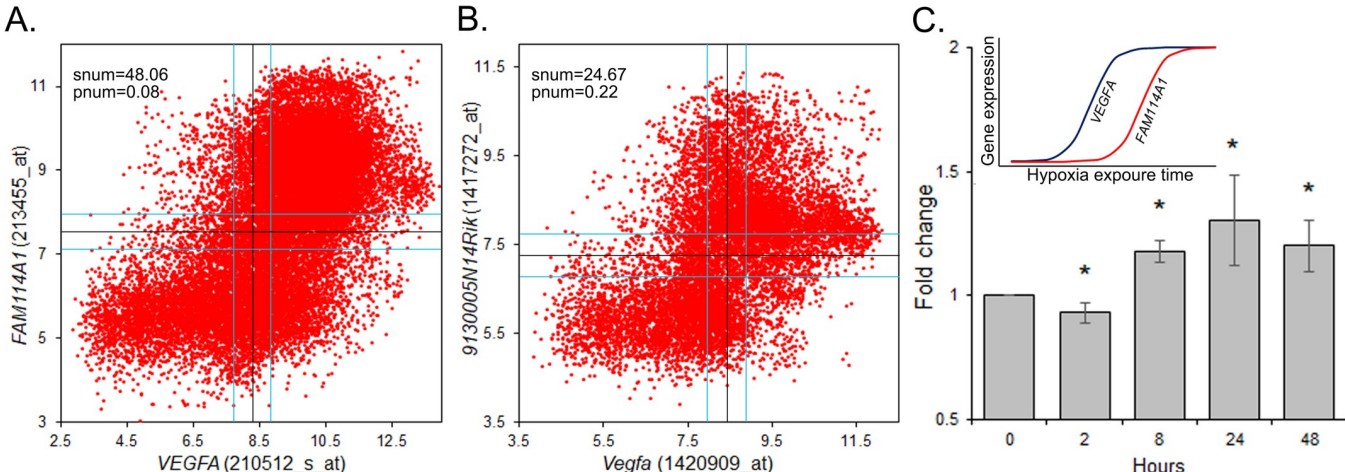

**Fig 2. Identification of *FAM114A1* as a novel hypoxia responsive gene.** (a) Scatter plot depicting the expression profile of *VEGFA* (probe ID, 210512_s_at; x-axis) and *FAM114A1* (probe ID, 226697_at; y-axis) in n = 25,955 'human' samples. (b) Scatter plot depicting the expression profile of *Vegfa* (probe ID, 1420909_at; x-axis) and *Fam114a1* (9130005-N14Rik, probe ID, 1417272_at; y-axis) in n = 11,758 'mice' samples (Correlation = 0.451606). (c) RT-PCR results depicting expression level of *FAM114A1* under normoxia (21% $O_2$) and at 2, 8, 24 and 48 hours of constant 1% $O_2$ measured in HPAEC. Insert, gene expression dynamics of *VEGFA* and *FAM114A1* along the boolean path. (*, P<0.05).

i.e., 1.18, 1.3 and 1.4 fold increase, when compared to the *VEGFA* where the fold change is 4.8, 8.8 and 8.3 fold at 8, 24 and 48 hours respectively. Interestingly, at around 2 hours of 1% $O_2$ treatment the *FAM114A1* levels are significantly low (P<0.05; Fig 2C).

## Distinctive properties of *FAM114A1* and its role in hypoxia

Although the gene was first identified in the nervous system (Noxp20), [13] its expression is low in the nervous system (Fig 3A). Therefore, in order to better understand the role of *FAM114A1*, particularly under hypoxic environment, we systematically examined its distinctive properties.

Sequence analyses of *FAM114A1* indicate numerous hypoxia-response elements (HREs, 5′-RCGTG-3′) [14] that are located in the promoter and 5' untranslated region (Fig 3B). Additionally, intron-1 has five HREs of which four are part of an independently transcribed intronic miRNA, miR-574 (at positions chr4:38,868,032–38,868,127, GRCh38/hg38; Fig 3B). Further, the region containing miR-574 is evolutionarily conserved (Fig 3B) and since miR-574-3p is reported to reduce *VEGFA* [15], we believe that there is a functional link between *FAM114A1* and *VEGFA*.

The gene network estimation analysis, using GeneMANIA, reveals *FAM114A1* having a shared protein domain with its homologue *FAM114A2* (Fig 3C). The top 20 *FAM114A1* interacting genes include *KDELR2*, *KDELR3*, *P4HA2* and *PON2* which are part of the broader list of genes having Boolean implication 'low = > low' with *VEGFA*. Beside this, the interaction analysis reveals *AKT1S1* (AKT1 substrate 1 or proline-rich Akt substrate of 40 kDa (PRAS40)) and *N4BP1* (NEDD4 binding protein), as the two most physically interacting genes (Fig 3C).

## Discussion

In the present study, we used a Boolean implication relationship on publicly available microarray datasets and identified *FAM114A1* (Family with Sequence Similarity 114 Member A1) as a novel hypoxia related gene (Fig 2). Since the computational approach is reliant on the dynamic range of expression of the seed gene, we used *VEGFA* as seed gene due to its robust hypoxia

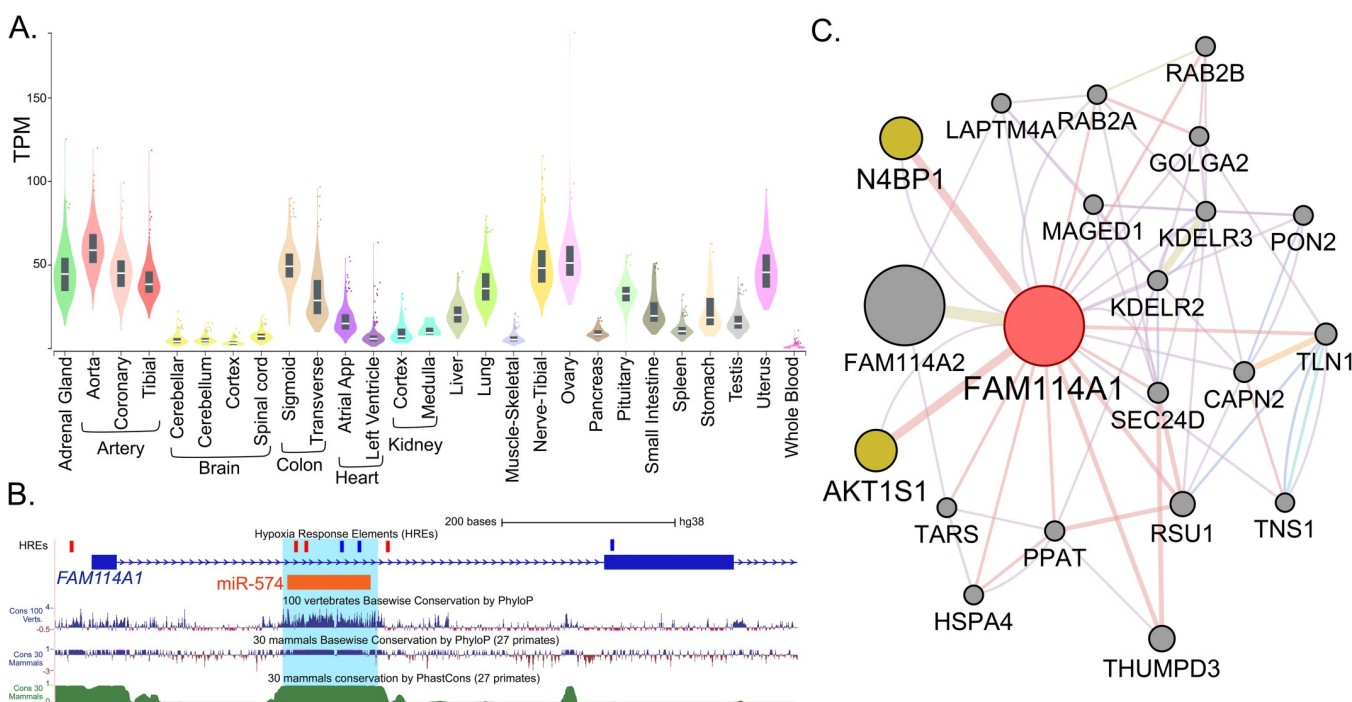

**Fig 3. Distinctive properties of *FAM114A1* and its role in hypoxia.** (a) *FAM114A1* is expressed in most of the tissues but its expression is low in the nervous system. (b) Position of hypoxia-response elements (HREs) consenses in *FAM114A1* and intronic miRNA, miR-574 in intron-1. The region constituting miR-574 is evolutionarily conserved. (c) Network analysis reveals physical interaction of *FAM114A1* with *AKT1S1* (AKT1 substrate 1) and *N4BP1* (NEDD4 binding protein 1).

sensitivity [16], and checked the Boolean implications of *VEGFA* to the other genes. We observed previously known hypoxia-related gene (>85% hypoxia related genes), which is important since such findings validate to a degree our novel findings [11]. The two important advantages of this approach a) is the coverage of a wide range of samples from a publicly available datasets of any conventional experiments and b) this method can identify important genes that are functionally conserved in humans and mice.

The Boolean implication '*VEGFA* low⇒*FAM114A1* low' enabled us to hypothesize a relationship at a molecular level, particularly in certain specific cell types. As expected, we discovered a putative relationship of similar pattern i.e., simultaneous upregulation of both *VEGFA* and *FAM114A1* at 8 hours of 1% $O_2$ (Fig 2). We also uncovered similar responses in some of the previous studies [17, 18]. For example, both genes were significantly upregulated in HPAECs and cardiac microvascular endothelial cells treated with 1% $O_2$ (S2 Fig) [17]. Importantly, the hypoxia-induced upregulation was not immediate, much like in previous studies [17]. Similar responses were also revealed in lymphatic endothelial cells cultured in 1% $O_2$ for 24 hours [19] and Burkitt's lymphoma cell line P493-6 cultured at 0.1% $O_2$ for 29 hours [18], all having common cell lineage. By contrast, *FAM114A1* fails to respond in a similar way when endothelial cells are exogenously treated with VEGF (S3A and S3B Fig) [20]. Further, the relationship was absent in human primary renal proximal tubule epithelial cells treated with 1% $O_2$ for 24 h (S3C and S3D Fig) [21], indicating an endothelial cell type-dependent response. It is worth noting that most of these observations were not highlighted in the previous studies. For example, the fold change of *FAM114A1* was not among the top 25 genes with >4 fold change as reported in Kim et al., (S3E and S3F Fig) [18], or the differences are revealed only after we normalized Affymetrix platform using RMA (Robust Multichip Average) [11, 17, 22, 23].

Additional evidence that supports and strengthens our discovery of *FAM114A1* as a hypoxia-related gene includes a) the presence of numerous HREs in its promoter and 5' untranslated region (Fig 3B), which are hypoxia-inducible factor (HIF) binding site that lead to hypoxia-induced transcriptional response in the target genes [24, 25] and b) the presence of an evolutionarily conserved and functionally related miRNA-574 embedded in the intron-1 of *FAM114A1*, miRNA-574, which negatively regulates *VEGFA* translation (Fig 3B) [15].

To distinguish whether *FAM114A1* upregulation is a hypoxia related or predominantly related to *VEGFA* activation, we explored into conditions involving non-hypoxia related *VEGFA* activation such as in cell growth, apoptosis, cell proliferation and tumor development [26]. A similar relationship was observed during neuronal differentiation of human SH-SY5Y neuronal cells (S4A and S4B Fig) and in methotrexate sensitive (low) vs resistant (high) HT29 colon cancer cells (S4C and S4D Fig) [27, 28]. A recent study report an increase *FAM114A1* expression in the failing heart [29], which we believe could be related to *VEGFA* [30]. However, *VEGFA* expression does not change when *FAM114A1* is upregulated in breast cell line subpopulation (MDA-MB-231) (S4E and S3F Fig) [31]. Overall, these results indicate that the interaction between *VEGFA* and *FAM114A1* is typically maintained in endothelial or closely related cell lineages when exposed to hypoxic conditions but seldom in non-endothelial and non-hypoxic conditions.

When we explored previous studies on *FAM114A1* to understand its function, particularly under hypoxia, we found that apart from its role in neuronal cell development [13], melanocyte apoptosis [32] and an association with ankylosing spondylitis [33], there was very little known about this gene or its homologue *FAM114A2*. A recent study in *Fam114a1*$^{-/-}$ mice showed that cardiac function was markedly restored in the knockout mice, in angiotensin II infusion model of hypertension, when compared to the controls [29], indicating its role in oxidative stress. Although, this study reported that *FAM114A1* regulated the expression of angiotensin type 1 receptor (*AGT1R*), there was not an apparent link between the two genes in the conventional network analysis (Fig 3C). Interestingly, we observed a Boolean implication *FAM114A1* low = > *AGTR1* low (S5 Fig), clearly highlighting the importance of our approach. On further exploring *AGTR1* related pathway, i.e., Renin Angiontensin system, we noticed that *FAM114A1* has a Boolean relationship with *ACE2* (angiotensin-converting enzyme 2 also a functional receptor on cell surfaces through which SARS-CoV-2 enters the host cells) and *AGT* (angiotensinogen) but not with *ACE* (angiotensin-converting enzyme) and *AGTR2* (angiotensin type 2 receptor) (S5 Fig). This is highly informative because a Boolean implications of *FAM114A1* with *ACE2* and *AGTR1* and 'no relation' with the leading candidate of RAS i.e., *ACE*, would depict its preferential interaction or regulation of RAS, especially when *ACE* and *ACE2* has opposing role in the same pathway [34, 35]. Additionally, both *ACE2* and *AGTR1* are upregulated in hypoxia [36, 37] and both has role in failing human heart [29, 38].

One of the previous study labelled the protein product of this gene as 'nervous system overexpressed protein 20' (*NOXP20*), detected from *in situ hybridization* of mouse cryostat sections [13]. However, the transcript levels in different tissue-wide expression datasets e.g., GTEx, Illumina bodyMap2 transcriptome etc., (Fig 3A) is lowest in the nervous system (brain and spinal cord) when compared to all the other tissues [39, 40]. We therefore call this gene *OXSI1* (*oxidative stress induced 1*), representing the noticeable characteristic from ours and numerous other studies. The lack of any supporting literature led us to focus on gene network estimation, which indicates that *FAM114A1* closely interact with *AKT1S1* (PRAS40), a subunit of mTORC1, and *N4BP1*. In the context to our current finding, *AKT1S1* is known to be activated by hypoxia [41]. Its activation in the nerve cells protect neuronal cell from damage [41] and in endothelial cells, it suppresses atherogenesis [42], both through inhibition of mTORC1 signaling. Interestingly, both *AKT1S1* and *N4BP1* are known to oppositely regulate NF-κB

transcriptional activity. While *N4BP1* inhibits TLR-dependent (Toll-Like Receptor-dependent) activation of NF-κB by interacting with IκB kinase γ [43, 44], *AKT1S1* promotes NF-κB transcriptional activity through its association with p65 [45]. While the activation of NF-κB in hypoxia is well known, the several proposed mechanisms still needs consolidation [46–49]. Additionally, the genomic proximity of *FAM114A1* with some of the *TLR* genes i.e., *TLR1*, *TLR6* and *TLR10*, specifically *TLR6*, which has an overlapping promoter region but transcribing in opposite direction, may indicate that genes in proximity are driven by their shared biological function [50]. After appreciating the fact that *FAM114A1* interacts with two oppositely regulating entities of NF-κB one can anticipate the critical role it may play under different conditions. Therefore, future studies on *FAM114A1*, including its potential role in NF-κB signaling through its interaction with *ATK1S1* and/or *N4BP1*, is critical.

Overall, we provide a computational method for identifying hypoxia related gene that bypasses the conventional approach that are time consuming and costly. Since the method utilizes expression data from thousands of diverse samples, it holds the potential to reveal novel candidate markers. Our result suggest that *FAM114A1* is a hypoxia related gene, with a role in oxidative stress and several additional evidences, including it hosting a hypoxia related miRNA, support this observation. However, due to the lack of sufficient literature on *FAM114A1*, particularly our findings on the network interaction with NF-κB, it will be critical to further investigate mechanisms underlying its activation and its implications in physiology and pathophysiology.

## Methods

### Boolean implications analysis

We utilized more than 25,955 publicly available human dataset to perform Boolean implication analysis. The seed gene was *VEGFA* (vascular endothelial growth factor A, Gene ID: 7422, Fig 1A) which is a well-established hypoxia responsive/sensitive gene. We explored *ANGPTL4*, *PPARG*, *PTGIS* and *INHBA* as potential seed genes, one at a time, because of their higher fold change in HPAECs, as seen in previous microarray study, when exposed to hypoxia [51]. However, since the basal expression of these genes was low in most of the tissues (S6 Fig), we chose *VEGFA* as a more appropriate seed gene for hypoxia related gene discovery, especially when using endothelial cells for further validation. Interestingly, when we individually used *ANGPTL4*, *PPARG*, *PTGIS* and *INHBA* as the seed genes, *FAM114A1* was the only common gene in the top 100 Boolean related genes listed for each seed gene (S7 Fig).

Out of the four probesets targeting *VEGFA* i.e., 210512_s_at, 210513_s_at, 211527_x_at and 212171_x_at, we used probeset 210512_s_at in our primary analysis because of its robust signal. The software MiDReG (mining developmentally regulated genes) which couples gene expression patterns with "if-then" rules (Boolean implications) was used to predict functionally important *VEGFA*/hypoxia-related genes [11, 12]. The BooleanNet statistical threshold for this analysis was set at SThr > 40 and pThr < 0.2. At this stringent cutoff threshold we identified 310 probesets (230 genes) that has Boolean relation with *VEGFA*. We then use multilayer filters: 1) inclusion of strong relationship with *VEGFA*, 2) a similar relationship with the other three probesets of *VEGFA*, 3) conserve relationship in mice and 4) the candidate gene is not reported previously as a hypoxia related genes. The selected gene/s were proceeded for *in-vitro* validation.

### Cell culture

Primary Pulmonary Artery Endothelial Cells (HPAEC) was purchased from ATCC (PCS-100-022™). For maintaining normal growth, we followed protocol as indicated by ATCC. The

Endothelial Cell Growth Kit-BBE (ATCC® PCS-100-040) was added as indicated. We passaged the cells when cultures reached approximately 80% confluence. For the hypoxia experiments, equal numbers of cells were plated in five 60mm cell culture dishes and maintained in regular incubator of room air, 5% $CO_2$ and 37˚C. On day three of cell expansion i.e., ~70% confluence, four plates were transferred to an incubator with 1% $O_2$, 5% $CO_2$ and 37˚C. Cells from each dish were used to isolate RNA at 0, 2, 8, 24 and 48 hours of hypoxia exposure. In order to get a robust reproducible readout the technical replicates used for RT-PCR are from three different passages.

### Real-Time qRT-PCR

RNA was isolated from tissue samples using RNeasy Mini Kit (Qiagen, US). We used SYBR® Green Master Mix for RT-PCR which is a pre-formulated, optimized, universal 2X master mix for real-time PCR workflows. RT-PCR was performed on CFX96 Real-Time PCR System. The specific primers used for RT-PCR are listed in S3 Table. The relative transcript levels in hypoxia is compared to its levels in normoxia after normalizing to a housekeeping gene i.e., *GAPDH* (S3 Table).

### Network estimation analysis

To predict the molecular function of *FAM114A1* we used geneMANIA (http://genemania.org) on Cytospace program (v3.4.0). This helps us identify various types of interactions with the other genes in the network as it uses a very large set of functional association data [9] to find out other genes that are related to *FAM114A1*. We used the default values for this query i.e., maximum resultant genes = 20 and maximum resultant attributes = 10. The information on physical and genetic interactions, pathways, co-expression, co-localization and protein domain similarity were obtained as a readout.

### Statistical analysis

The data are shown as means ± standard errors (SEM). Paired or unpaired Student's *t*-test and one-way analysis of variance (ANOVA) with Bonferroni multiple comparison test were used for statistical analysis. A *p* value <0.05 was considered as statistically significant.

### Supporting information

**S1 Fig. Expression profile of *HIF1A* in HPAEC under normoxia (21% $O_2$) i.e., at 0 hour and at 2, 8, 24 and 48 hours of constant 1% $O_2$.**
(PDF)

**S2 Fig. Expression profile of *VEGFA* and *FAM114A1* in the data extracted from Costello et al., 2008 (GEO accession: GSE12792, PMID: 18469115).** Expression profile of *VEGFA* and *FAM114A1* in the pulmonary microvascular endothelial cells (a, b) and cardiac microvascular endothelium (c, d) under normoxia (21% $O_2$) and at 3, 24 and 48 hours of constant 1% $O_2$ as reported in Costello et al., *Am J Physiol Lung Cell Mol Physiol* 2008. (*, P<0.05 when compared to the normoxia).
(PDF)

**S3 Fig. Hypoxia-induced changes in the expression profile of *VEGFA* and *FAM114A1* in different cell lineage from previously reported data.** (a, b) VEGF-treated endothelial cells treated for 0, 2 and 4 hours (GEO accession: GSE18913; PMID: 19965691). (c, d) Human renal proximal tubule epithelial cells (RPTEC, obtained from Lonza) exposed to 1% oxygen for 24 h.

(GEO accession: GSE12792; PMID: 18984585). (e, f) P493-6 cells (Human Lymphoblastoid Cell Line) incubated in normoxic (20% $O_2$) or hypoxic condition (0.1% $O_2$) for 29 hr. (GEO accession: GSE4086; PMID: 16517405). *, P<0.05 when compared to the normoxia.
(PDF)

**S4 Fig. Non-hypoxia-related changes in the expression profile of *VEGFA* and *FAM114A1*.** (a, b) Similar changes in *VEGFA* and *FAM114A1* in the undifferentiated neuroblastoma cells (SH-SY5Y), i.e. before PMA induced differentiation (UD), 48 hours differentiated and untransfected cells (D-UT), 48 hours differentiated and MeCP2 decoy transfected cells (D-MD) and 48 hours differentiated and Control decoy transfected cells (D-CD) (GEO accession: GSE4600; PMID: 16682435). (c, d) Comparison of human HT29 colon cancer cells that are sensitive and resistant to methotrexate (GEO accession: GSE11440; PMID: 18694510). (e, f) CXCR4-positive subpopulation (expressing CXCR4), CXCR4-positive subpopulation treated with SDF-1 (CXCR4 treated with SDF-1alpha, ligand for CXCR4, also called CXCL12) for one hour and CXCR4-negative subpopulation (not expressing CXCR4) (GEO accession: GSE15893; PMID: 20603605). *, P<0.05.
(PDF)

**S5 Fig. Boolean relationships between *FAM114A1* and candidate genes from Renin Angiotensin system (RAS) taken from the Affymetrix Human U133 Plus 2.0 dataset.** At SThr = 10 and pThr = 0.1 no relation between *FAM114A1* and *ACE* (a), *FAM114A1* low = > *ACE2* low (b), *FAM114A1* low = > *AGTR1* low (c), no relation for *FAM114A1* vs *AGTR2* (d), no relation for *FAM114A1* vs *AGT* (e) and *FAM114A1* low = > *REN* low (f).
(PDF)

**S6 Fig. Basal expression of *VEGFA*, *ANGPTL4*, *PPARG*, *PTGIS* and *INHBA* in different tissues as indicated in GTEx database.** Y-axis indicates the median TPM (Transcript Per Million).
(PDF)

**S7 Fig. Venn diagram indicating the common shared genes (among the top 100 Boolean related genes) when *ANGPTL4*, *PPARG*, *PTGIS* and *INHBA*, respectively, are used seed genes one at a time.** *FAM114A1* (red circle) is the common for all seed genes.
(PDF)

**S1 Table. The probesets (genes) detected by MiDReG that pairs with *VEGFA* at a Boolean-Net statistical threshold of SThr > 40 and pThr < 0.2.**
(XLSX)

**S2 Table. Top probesets (genes) that have a strong relationship with *VEGFA*.**
(XLSX)

**S3 Table. Fold change in the expression levels of candidate genes at 2, 8, 24 and 48 hours of 1% $O_2$ treatment compared to its baseline expression levels at 21% $O_2$.**
(XLSX)

## Author Contributions

**Conceptualization:** Tsering Stobdan, Debashis Sahoo, Gabriel G. Haddad.

**Data curation:** Tsering Stobdan, Debashis Sahoo.

**Formal analysis:** Tsering Stobdan.

**Funding acquisition:** Gabriel G. Haddad.

**Investigation:** Tsering Stobdan, Gabriel G. Haddad.

**Methodology:** Tsering Stobdan, Debashis Sahoo.

**Project administration:** Gabriel G. Haddad.

**Resources:** Gabriel G. Haddad.

**Software:** Debashis Sahoo.

**Supervision:** Debashis Sahoo, Gabriel G. Haddad.

**Validation:** Tsering Stobdan.

**Visualization:** Tsering Stobdan, Gabriel G. Haddad.

**Writing – original draft:** Tsering Stobdan.

**Writing – review & editing:** Tsering Stobdan, Gabriel G. Haddad.

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
