## [Decision Letter · Decision Letter 0]

19 Jul 2022

PONE-D-22-15421A Boolean approach for novel hypoxia-related gene discoveryPLOS ONE

Dear Dr. Haddad,

Thank you for submitting your manuscript to PLOS ONE. After careful consideration, we feel that it has merit but does not fully meet PLOS ONE’s publication criteria as it currently stands. Therefore, we invite you to submit a revised version of the manuscript that addresses the points raised during the review process.

We look forward to receiving your revised manuscript.

Kind regards,

Junming Yue

Academic Editor

PLOS ONE

Journal Requirements:

This work was supported by NIH grant (R01 HL127403-04) GGH.

4. PLOS requires an ORCID iD for the corresponding author in Editorial Manager on papers submitted after December 6th, 2016. Please ensure that you have an ORCID iD and that it is validated in Editorial Manager. To do this, go to ‘Update my Information’ (in the upper left-hand corner of the main menu), and click on the Fetch/Validate link next to the ORCID field. This will take you to the ORCID site and allow you to create a new iD or authenticate a pre-existing iD in Editorial Manager. Please see the following video for instructions on linking an ORCID iD to your Editorial Manager account: https://www.youtube.com/watch?v=_xcclfuvtxQ.

6.  We note your current data statement: 'Data Availability: Yes - all data are fully available without restriction

All relevant data are within the manuscript and its Supporting Information files.'

And the following is stated in the paper: "...utilized more than 25,955 publicly available human dataset to perform Boolean implication analysis".

Please confirm the stated datasets are included in paper and/or the uploaded supporting information.

Reviewers' comments:

Reviewer's Responses to Questions

**Comments to the Author**

1. Is the manuscript technically sound, and do the data support the conclusions?

Reviewer #1: Partly

2. Has the statistical analysis been performed appropriately and rigorously? 

Reviewer #1: Yes

3. Have the authors made all data underlying the findings in their manuscript fully available?

Reviewer #1: Yes

4. Is the manuscript presented in an intelligible fashion and written in standard English?

Reviewer #1: Yes

5. Review Comments to the Author

Reviewer #1: The manuscript by Tsering Stobdan et. al. used a Boolean method to identify the hypoxia-responsive genes from online database. The story is simple and interesting; however the manuscript is too primitive and need more work on it before it can be considered for publication.

1. Some information in the parentheses need to be clarified, such as ‘potential therapeutic targets (PAHS-032Z array (Qiagen))’ in the first and second paragraphs of Introduction; ‘it is clear that identification of new or next generation of targets will require data from diverse samples types and enormous sample numbers and a newer method to analyze (Ref).’ in the second paragraph of Introduction.

2. Fig. 1, no error bars in all the 0 h groups. Fig 1I, the induction of ITGAV at 8h and 48 h are too modest, and it is hard to believe the significance.

3. The result of this Boolean method seems depend on the seed gene, have the authors ever tried use another seed gene such as EPO and compare the results from different seed genes?

4. The citations of Fig. S3, Fig. S4 in manuscript do not match the figure legend.

5. Cannot find the figure related with this sentence, ‘VEGFA expression does not change when FAM114A1 is upregulated in breast cell line subpopulation (MDA-MB-231) (Fig S4c)’.

6. PLOS authors have the option to publish the peer review history of their article (what does this mean?). If published, this will include your full peer review and any attached files.

Reviewer #1: **Yes: **Wenjing Zhang

---

## [Author Response · Author response to Decision Letter 0]

28 Jul 2022

Reviewer #1: The manuscript by Tsering Stobdan et. al. used a Boolean method to identify the hypoxia-responsive genes from online database. The story is simple and interesting; however the manuscript is too primitive and need more work on it before it can be considered for publication.

Response: We agree with the reviewer about the simplicity of the approach. Yet it is a powerful method and this is only possible after we were able to combine a normalized expression of 25,955 human microarray data (obtained from Gene Expression Omnibus (GEO)) in a single database. We also pooled 11,758 mouse microarray data for additional validation. The database is available online (http://hegemon.ucsd.edu/Tools/explore.php?key=global).

1. Some information in the parentheses need to be clarified, such as ‘potential therapeutic targets (PAHS-032Z array (Qiagen))’ in the first and second paragraphs of Introduction; ‘it is clear that identification of new or next generation of targets will require data from diverse samples types and enormous sample numbers and a newer method to analyze (Ref).’ in the second paragraph of Introduction.

Response: Thank you. We have updated these parts with appropriate references. 

2. Fig. 1, no error bars in all the 0 h groups. Fig 1I, the induction of ITGAV at 8h and 48 h are too modest, and it is hard to believe the significance.

Response: Since the bar plots are presented as fold change relative to the 0-hour (room air/21% O2) expression, the expression level at 0-hour is set to a baseline value of ‘1’. 

For the modest ITGAV levels at 8h and 48h, the average values are close and there is a very small standard deviation. A t-test depicts the differences as statistically significant. We have now added a supplementary table i.e., Table S3, which has the values.

3. The result of this Boolean method seems depend on the seed gene, have the authors ever tried use another seed gene such as EPO and compare the results from different seed genes?

Response: We explored ANGPTL4, PPARG, PTGIS and INHBA as potential seed genes, one at a time, because of their higher fold change in HPAECs (microarray platform) when exposed to hypoxia (Table 1 from Manalo et al. Blood. 2005). However, since the basal expression of these genes was low (including EPO) in most of the tissues (Figure S6), we chose VEGFA as a more appropriate seed gene for hypoxia related gene discovery, especially when using endothelial cells for further validation.

Interestingly, recently, in response to the Reviewer’s query, when we individually used ANGPTL4, PPARG, PTGIS and INHBA as the seed genes, FAM114A1 was the only common gene in the top 100 Boolean related genes listed for each seed gene (Figure S7).

We thank the reviewer for this important comment and we have now added this information in the method section of the manuscript.

Figure S6: Basal expression of VEGFA, ANGPTL4, PPARG, PTGIS and INHBA in different tissues as indicated in GTEx database. TPM, Transcript Per Million.

Figure S7: Venn diagram indicating the common shared genes (among the top 100 Boolean related genes) when ANGPTL4, PPARG, PTGIS and INHBA, respectively, are used seed genes one at a time. FAM114A1 (red circle) is the common for all seed genes.

4. The citations of Fig. S3, Fig. S4 in manuscript do not match the figure legend.

Response: Corrected.

5. Cannot find the figure related with this sentence, ‘VEGFA expression does not change when FAM114A1 is upregulated in breast cell line subpopulation (MDA-MB-231) (Fig S4c)’.

Response: This is on Figure S4e – f. Additional details are added to its figure legend. Briefly, the cells (MDA-MB-231) with CXCR4-positive subpopulation (expressing CXCR4) have high level of FAM114A1 when compared with the other two groups i.e., cells with CXCR4-positive subpopulation treated with SDF-1 (CXCR4 treated with SDF-1alpha), ligand for CXCR4, also called CXCL12) for one hour and CXCR4-negative subpopulation (not expressing CXCR4).

6. PLOS authors have the option to publish the peer review history of their article (what does this mean?). If published, this will include your full peer review and any attached files.

Response: OK

---

## [Decision Letter · Decision Letter 1]

10 Aug 2022

A Boolean approach for novel hypoxia-related gene discovery

PONE-D-22-15421R1

Dear Dr. Haddad,

We’re pleased to inform you that your manuscript has been judged scientifically suitable for publication and will be formally accepted for publication once it meets all outstanding technical requirements.

Kind regards,

Junming Yue

Academic Editor

PLOS ONE

Additional Editor Comments (optional):

Reviewers' comments:

Reviewer's Responses to Questions

**Comments to the Author**

1. If the authors have adequately addressed your comments raised in a previous round of review and you feel that this manuscript is now acceptable for publication, you may indicate that here to bypass the “Comments to the Author” section, enter your conflict of interest statement in the “Confidential to Editor” section, and submit your "Accept" recommendation.

Reviewer #1: All comments have been addressed

2. Is the manuscript technically sound, and do the data support the conclusions?

Reviewer #1: Yes

3. Has the statistical analysis been performed appropriately and rigorously? 

Reviewer #1: Yes

4. Have the authors made all data underlying the findings in their manuscript fully available?

Reviewer #1: Yes

5. Is the manuscript presented in an intelligible fashion and written in standard English?

Reviewer #1: Yes

6. Review Comments to the Author

Reviewer #1: (No Response)

7. PLOS authors have the option to publish the peer review history of their article (what does this mean?). If published, this will include your full peer review and any attached files.

Reviewer #1: **Yes: **Wenjing Zhang

---

## [Editor Report · Acceptance letter]

14 Aug 2022

PONE-D-22-15421R1 

A Boolean approach for novel hypoxia-related gene discovery 

Dear Dr. Haddad:

I'm pleased to inform you that your manuscript has been deemed suitable for publication in PLOS ONE. Congratulations! Your manuscript is now with our production department. 

Kind regards, 

on behalf of

Dr. Junming Yue 

Academic Editor

PLOS ONE